# Opportunistic evaluation of modelled sea ice drift using passively drifting telemetry collars in Hudson Bay, Canada

Ron R. Togunov[1], Natasha J. Klappstein[2], Nicholas J. Lunn[3], Andrew E. Derocher[2], Marie Auger-Méthé[4]

[1]Department of Zoology, Institute for the Oceans and Fisheries, University of British Columbia, Vancouver, BC V6T 1Z4, Canada
[2]Department of Biological Sciences, University of Alberta, Edmonton, AB T6G 2E9, Canada
[3]Wildlife Research Division, Science & Technology Branch, Environment and Climate Change Canada, CW-422 Department of Biological Sciences, University of Alberta, Edmonton, AB T6G 2E9, Canada
[4]Department of Statistics, Institute for the Oceans and Fisheries, University of British Columbia, Vancouver, BC V6T 1Z4, Canada

*Correspondence to*: Ron R. Togunov (r.togunov@oceans.ubc.ca)

**Abstract**

Sea ice drift plays a central role in the Arctic climate and ecology through its effects on the ice cover, thermodynamics, and energetics of northern marine ecosystems. Due to the challenges of accessing the Arctic, remote sensing has been used to obtain large-scale longitudinal data. These data are often associated with errors and biases that must be considered when incorporated into research. However, obtaining reference data for validation is often prohibitively expensive or practically unfeasible. We used the motion of 20 passively drifting high-accuracy GPS telemetry collars originally deployed on polar bears, *Ursus maritimus*, in western Hudson Bay, Canada to validate a widely used sea ice drift dataset produced by the National Snow and Ice Data Centre (NSIDC). Our results showed that the NSIDC model tended to underestimate the 'horizontal' and 'vertical' (i.e. 'u' and 'v') components of drift. Consequently, the NSIDC model underestimated magnitude of drift, particularly at high ice speeds. Modelled drift direction was unbiased; however, it was less precise at lower drift speeds. Research using these drift data should consider integrating these biases into their analyses, particularly where absolute ground speed or direction is necessary. Further investigation is required into the sources of error, particularly in under-examined areas without in situ data.

## 1 Introduction

Many research fields increasingly depend on remote sensing to collect environmental data. The raw data from various remote sensing sources are often combined using modelling and interpolation techniques to create an accessible gridded product (Reichle, 2008). For example, the Hadley Centre Sea Ice and Sea Surface Temperature data set, which combines data from numerous sources including active and passive satellite sensors, ice charts, and historic records (Titchner and Rayner, 2014). However, measurement errors and assimilation biases can lead to large inaccuracies (Reichle, 2008). If the degree of measurement error is greater than the variability of the system being modelled, it could lead to spurious results (Auger-Méthé et al., 2016b). Quantifying error in remotely sensed data can be used to improve these data products (Cressie et al., 2009), and is important for data assimilation and the development of new products (Meier et al., 2000; Sumata et al., 2014, 2015a). However, assessing these errors is challenging, particularly in remote areas that are difficult to ground truth.

Sea ice studies often rely on remotely sensed data due to the remote, vast, and dynamic nature of the environment. Sea ice drift is a fundamental contributor to the dynamism of the Arctic ecosystem. Ice drift affects important thermodynamic processes through the formation of polynyas and leads (Marcq and Weiss, 2012), modulates ice deformation rates (Bouillon and Rampal, 2015; Rampal et al., 2009), and can determine spatial distribution and configuration of different ice ages and thicknesses (Hutchings and Rigor, 2012; Mahoney et al., 2019). It also drives the rate of sea ice export, which affects ice extent throughout the Arctic (Rampal et al., 2009). Therefore, ice drift is often considered in models of ice cover characteristics, overall sea ice mass throughout the Arctic, and global climate patterns (Hunke et al., 2010; Kimura and Wakatsuchi, 2000; Kwok et al., 2013). In addition to geographic and environmental studies, ice drift has received increased attention in ecological research. Ice drift influences the distribution and biomass of plankton (Hop and Pavlova, 2008; Kohlbach et al., 2017; Onodera et al., 2015; Thorpe et al., 2007), as well as polar bear (*Ursus maritimus*) behaviour and energetics (Auger-Méthé et al., 2016a; Durner et al., 2017; Mauritzen et al., 2003). In addition to its effects on geophysics and wildlife, ice drift is also important in describing transport of microplastics in the Arctic (Peeken et al., 2018). Given its broad application, the accuracy of ice drift data is critical when drawing geophysical and ecological conclusions.

Several sources of ice drift data are available at variable spatiotemporal resolutions (Sumata et al., 2014). Although the data and models used vary between ice products, ice drift estimates are generally estimated from combinations of buoy data, weather forecast models, and satellite measurements. These data sources vary in coverage, resolution, accuracy, and sensitivity to environmental/meteorological conditions and, therefore, result in products with variable sources of error (Mahoney et al., 2019; Sumata et al., 2014). In this paper, we sought to quantify these errors in a widely employed sea ice drift data product produced by the National Snow and Ice Data Center (NSIDC; Boulder, CO): Polar Pathfinder Daily 25 km EASE-Grid Sea Ice Motion Vectors (hereafter, NSDIC drift; Tschudi et al., 2019, 2020). NSIDC drift estimates are produced by assimilating drift obtained from several satellite-based sensors, buoys, and modelled wind fields, providing among the most extensive, high resolution, and complete spatial coverage. In addition, NSDIC drift product has the longest temporal coverage of any sea ice drift products extending from 1978 to the present (Tschudi et al., 2020).

Although research has examined the accuracy of older versions of NSIDC drift (e.g., Ruslan I. May, 2018; Schwegmann et al., 2011; Sumata et al., 2014, 2015b), the latest major release (version 4.0) has yet to be externally evaluated. The NSIDC drift model integrates the movement of buoys from the International Arctic Buoy Program (IABP; http://iabp.apl.washington.edu/), and are the highest weighted input source driving the NSIDC model (Sumata et al., 2015a). Regions without such in situ measurements are more susceptible to bias (Mahoney et al., 2019; Sumata et al., 2015a; Tschudi

et al., 2020), and are therefore particularly important to evaluate.

There are two types of data that can be used to cross-validate ice drift: (1) other telemetry-based estimators including moored Doppler-based velocity measures and other high resolution satellites (e.g., Advanced Very High Resolution Radiometer (AVHRR) or Synthetic Aperture Radar (SAR)), and (2) in situ drifters, including buoys, ships, and manned stations (Lavergne et al., 2016). Other satellite-based estimates are associated with their own estimation errors, and Doppler-based

validation represent only errors in the area in which they are moored (Rozman et al., 2011). Some studies used in situ drifters (e.g., drifting research stations or buoys) as reference data, however, they are consequently limited in spatial extent (Hwang, 2013; Rozman et al., 2011; Tschudi et al., 2010). Since there are few sources of in situ sea ice drift data, at least one study quantifying NSIDC drift accuracy used the same IABP data that are integrated into NSIDC model for validation, which may underestimate bias (e.g., Sumata et al., 2014). Further, IABP buoys have historically used ARGOS location estimates, which

have spatial errors up to tens of kilometres and may be unsuitable for validation of drift during the periods/areas they were deployed (Hwang, 2013).

In this paper, we evaluate the bias and precision (hereafter, collectively referred to as *accuracy*) of NSIDC drift data in Hudson Bay using an opportunistic and independent source of sea ice drift validation data. We compared modelled NSIDC drift to drifting GPS collars that were originally deployed on polar bears but dropped onto sea ice. There has been no study of

the accuracy of any sea ice drift model in Hudson Bay. In addition, the bay does not have any IABP buoys, which drive the NSIDC model and its performance. Our objectives were to quantify drift accuracy within three domains: drift speed, drift direction, and the orthogonal (horizontal, 'u', and vertical, 'v') components of the drift vectors. We also explored whether accuracy varied with the underlying drift speed, across months, or across years.

## 2 Methods

We fitted polar bears in western Hudson Bay, Canada with satellite-linked GPS collars (Telonics ®, Mesa, Arizona) in August and September of 2004-2015 (Figure 1). Procedures for animal capture and handling are described by Stirling et al. (1989) and were approved annually by the University of Alberta Animal Care and Use Committee for Biosciences and by the Environment and Climate Change Canada Western and Northern Region Animal Care Committee. Protocols were in accordance with the Canadian Council on Animal Care. Collars were programmed to obtain GPS fixes every 4 h The locations obtained have a

high-accuracy, with errors < 31 m (D'Eon et al., 2002). Although deployed with the purpose of studying polar bear behaviour and space use, some collars may slip off the bears, release early due to premature failure of the release mechanism, or the bear

may die while the collars continue to transmit locations. In these instances, the observed displacement of the collars represents the motion of sea ice. We identified drifting collars either through 'activity' sensors in the collars or by manually comparing the observed collar displacement with sea ice satellite imagery (Appendix A). To verify that manually identified drifting collars

were passively drifting and not on active bears, we compared accuracy metrics for speed, direction, and u and v (relative to the NSIDC drift projection, EPSG:3408) among activity sensor collars, manually identified passive collars, and collars on active bears. Detailed methods and results of this comparison are presented in the Appendix B.

We used the motion of the identified drifting collars (following date of inactivity/drop off; hereafter simply, collars) to quantify the accuracy and precision of NSIDC drift data. The NSIDC product provides daily estimates of sea ice drift derived

from buoy data, National Center for Environmental Prediction and National Center for Atmospheric Research reanalysis wind vectors, and several satellite sensors including AVHRR, the Advanced Microwave Scanning Radiometer for Earth Observing System (AMSR-E), Scanning Multichannel Microwave Radiometer (SMMR), and the Special Sensor Microwave Imager / Sounder (SSMI/S; Tschudi et al., 2019, 2020). To match the NSIDC product, collar locations were projected into the 25 km EASE-grid North (EPSG: 3408) projection used by NSIDC. NSIDC represents drift as movement between 12:00 UTC of

subsequent days. To match the NSIDC temporal resolution, we subsampled the collar locations to a 24 h resolution by retaining locations from 13:00 UTC, the closest collar location to 12:00 UTC. Next, we calculated drift vectors/components (i.e., speed, direction, u, and v), then removed any vectors from locations > 24 h apart. Next, we interpolated the NSIDC drift to the first location of each collar drift vector using inverse distance weight (inverse distance power set to three and maximum distance of 50 km) to match the fix location.

The summary statistics chosen to quantify drift accuracy can lead to incomplete or spurious conclusions (Volkov et al., 2017). For example, root mean square and standard errors convey the magnitude of the error, but not the direction. Correlation coefficients between model and reference data describe model precision, but not accuracy. Some studies investigated the accuracy of the orthogonal components of drift (i.e., u and v) individually; however, this does not convey the accuracy in speed and direction, which are emergent properties of both components. For example, if the biases of the

orthogonal components are equal and scale proportionally, then direction estimates remain accurate. Conversely, if the biases are negatively correlated, they may partially cancel and result in speed estimates more accurate than appear when examining the drift components independently. Thus, in addition to the orthogonal u and v components of drift, we also quantified the accuracy of drift speed and direction.

We tested five key questions: (1) are the estimated model speeds significantly different from the collar speeds, (2) is

the relative speed accuracy dependant on the underlying drift speed being estimated, (3) are the estimated model directions significantly different from the collar directions, (4) is the direction accuracy dependant on the underlying drift speed, and (5) do the relationships between the model u (v) and collar u (v) components diverge significantly from 1. Because the data is spatiotemporally autocorrelated, with subsequent days having similar drift speeds and different collars sampling different regions of Hudson Bay, we could not use a simple paired t-test for the absolute speed bias (1). Instead, we used an intercept-

only generalized linear mixed effect model (GLMM; with a Gaussian error distribution) with absolute speed bias

($Speed_{NSIDC} - Speed_{collar}$) as the response, wherein a significant intercept represents a significant difference between the model and the collar speeds. To account for repeat sampling from different collars representing different regions, collar identity was used as a random effect. To account for temporal autocorrelation, we fit the model with a first-order autoregressive error process (AR1). For speed-dependant accuracy of model speed (2), we defined *relative speed accuracy* as the quotient of

NSIDC drift speed over collar speed, $\frac{Speed_{NSIDC}}{Speed_{collar}}$, with values > 1 representing overestimation and values < 1 representing underestimation. This *relative speed accuracy* was modelled as a function of $log(Speed_{collar})$ using GLMMs with gamma error distribution and a log-link function. We log transformed $Speed_{collar}$ because it is zero-bound and the relative difference in speed (and thus its relative effect on model accuracy) decays exponentially with increasing values. We used the same random effect and AR1 structure as in (1). We assessed the accuracy of model direction, $Direction_{NSIDC} - Direction_{collar}$, (3) using

a Watson-Williams test for homogeneity of means for circular data. Although this test does not incorporate autocorrelation, the absolute direction accuracy did not exhibit temporal autocorrelation (Figure 2). For the speed-specific direction accuracy (4), we defined *relative direction accuracy* as the linearized absolute difference in direction, $\tan\left(\frac{|Direction_{NSIDC} - Direction_{collar}|}{2}\right)$, where 0 represents model unanimity and departure from 0 represents increasing error. This *relative direction accuracy* was modelled as a function of $log(Speed_{collar})$ using the same GLMM procedures used for

testing speed-specific relative speed accuracy (2). Any differences in speed or direction between the NSIDC and collar drift ultimately emerge from the estimated u and v components of sea ice drift. We assessed the relationship between the orthogonal components of NSIDC and collar drift (5) using GLMM (with a Gaussian error distribution), with model u (v) modelled as functions of collar u (v), and the same random effect and AR1 structure as in (1), (2), and (4). All GLMMs were fit using penalized quasi-likelihood (GLMM$_{PQL}$; Breslow and Clayton, 1993) using the 'glmmPQL' function of the 'MASS' package

(Venables and Ripley, 2002). Using GLMM$_{PQL}$, enabled us to meet all our model criteria: non-linear models with random effects and an auto-regressive structure. As a broad metric of goodness of fit, we used a the GLMM$_{PQL}$ $R^2$ metric developed by Jaeger et al. (2017) using the 'r2beta' function in 'r2glmm' package. All data processing and analyses were conducted in R version 3.6.1 (R Core Team, 2019).

## 3 Results

We identified 20 drifting collars with locations from December-July of 2005-2015 (Figure 1 and Figure 3), with a mean of 520 ± 358 GPS fixes per collar (total = 10409). The largest number of identified collars in one year was in 2009 (n = 6). The motion for these six collars is depicted in the supplement video (http://doi.org/10.5446/45186), which depicts the large degree of concurrence of drift vectors across large spatial extent. After subsampling to a daily resolution, we analysed 1677 collar drift vectors. The number of drift vectors ranged from 71 vectors in July to 304 vectors in March (mean = 210 ± 83 vectors;

Figure 4).

## 3.1 Accuracy of NSIDC drift speed

Mean NSIDC drift speed was $5.8 \pm 4.5$ km d$^{-1}$ while mean collar speed was $8.4 \pm 7.1$ km d$^{-1}$, the difference in speed $Speed_{NSIDC} - Speed_{collar}$ was statistically significant (GLMM$_{PQL}$: intercept $\pm$ 95 % confidence interval (CI) = -3.0 $\pm$ 1.2 km d$^{-1}$, degrees of freedom (df) = 1657, t-value = -4.8, p-value < 0.0001; Figure 5). NSIDC drift speeds were slower than collar drift speeds in 63.1% of the vectors and only 10.4 % of NSIDC drift speeds were within $\pm$ 10 % of collar drift speeds (Figure 5a). The discrepancy in drift speed was more pronounced at higher collar drift speeds, with a significant relationship between the quotient ($\frac{Speed_{NSIDC}}{Speed_{collar}}$) and collar speed (GLMM$_{PQL}$: slope = -0.67, df = 1656, t-value$_{slope}$ = -38.80, p-value$_{slope}$ < 0.0001, R$^2$ = 0.53; Figure 5b). Collar drift speeds < 4.5 km d$^{-1}$ were overestimated by a median of 42 %, speeds between 4.5 and 9.0 km d$^{-1}$ were underestimated by a median of 26 %, and speeds > 9.0 km d$^{-1}$ were underestimated by a median of 51 % (Figure 5). There was intra-annual and inter-annual variation (based on 95% CIs) in the correlation of NSIDC drift speeds and collar drift speeds, however there was no apparent pattern (Figure 3 and Figure 4).

## 3.2 Accuracy of NSIDC drift direction

NSIDC drift directions were on average $2.6° \pm 53.9°$ left relative to the collar drift direction, although the mean difference was not significantly different from 0° (Watson-Williams test: df$_1$ = 1, df$_2$ = 1676, F-value = 0.003, p-value = 0.95; Figure 6 and Figure 7). Most (71.3 %) of the NSIDC drift directions were within $\pm$ 22.5° of the collar drift directions (Figure 7). NSIDC drift direction tended to be more accurate at higher collar drift speeds, with a significant relationship between relative direction accuracy and collar drift speeds (GLMM$_{PQL}$: slope = -0.83, df = 1656, t-value$_{slope}$ = -7.52, p-value$_{slope}$ < 0.0001, R$^2$ = 0.03; Figure 7).

## 3.3 Accuracy of orthogonal NSIDC drift components

Mean collar drift u component was $-0.9 \pm 7.7$ km d$^{-1}$ compared to $-0.7 \pm 4.3$ km d$^{-1}$ for NSIDC drift u drift. Mean collar drift v component was $-1.1 \pm 7.7$ km d$^{-1}$ compared to $-0.8 \pm 4.5$ km d$^{-1}$ for NSIDC drift v component drift. NSIDC and collar drift components were significantly related in both the u component (GLMM$_{PQL}$: slope $\pm$ 95 % CI = 0.38 $\pm$ 0.02, df = 1656, t-value$_{slope}$ = 37.58, p-value$_{slope}$ < 0.0001, R$^2$ = 0.46; Figure 8), and v component (GLMM$_{PQL}$: slope $\pm$ 95 % CI = 0.40 $\pm$ 0.02, df = 1656, t-value$_{slope}$ = 37.54, p-value$_{slope}$ < 0.0001, R$^2$ = 0.52; Figure 8). Although the components of NSIDC drift and collar drift were significantly correlated, the slopes of the regression were significantly underestimated (indicated by the slope estimate and 95 % CI being < 1).

**4 Discussion**

Using drifting collars as reference data for validation, we identified biases in the estimated speed and direction of NSIDC sea
ice drift model. NSIDC drift speeds tended to be underestimated, although drift direction was relatively accurate. This is due
to the underestimation of u and v components, which showed a similar magnitude in their bias. The biases in speed and
direction were related to the underlying drift speed as measured by the collars. NSIDC drift speeds tended to overestimate
slow collar drift (< 4.5 km h$^{-1}$) and underestimate high collar drift (> 4.5 km h$^{-1}$). This pattern is likely an effect of estimating
a zero-bound variable, and is consistent with other satellite-based sea ice drift products (Johansson and Berg, 2016; Mahoney
et al., 2019; Rozman et al., 2011; Sumata et al., 2014). As drift speeds approach 0 km d$^{-1}$, the probability of overestimation
approaches 1, and as drift speeds increase, the range of values that below the drift speed (i.e., underestimates) increases.
Although the bias is mathematically inevitable to some degree, the magnitude of the bias is not fixed and our results show that
the error can be high, with drift speeds underestimated by a median of 22.9% (1.4 km d$^{-1}$). This is similar to the drift bias
observed by Durner et al., (2017) in the Beaufort and Chukchi Seas, wherein mean daily model speed was underestimated by
a mean of 28.0% (2.25 km d$^{-1}$). These biases are small relative to the 25 km resolution of the satellite input data, however in
some analyses, the bias would compound over time. For example, cumulative/total daily drift calculated for 7 months
(corresponding to the months in which we obtained drift data) would be underestimated by > 295 km. Drift direction accuracy
increased at higher collar drift speeds. This is probably because magnitude and uniformity of sea ice displacement increase
with drift speed, and more likely to be detected by NSIDC's feature-matching algorithm (based on maximum cross-correlation;
Tschudi et al., 2019).

Our estimates of drift speed bias are greater than estimated in studies of NSIDC and other drift products (Durner et al., 2017;
Hwang, 2013; Johansson and Berg, 2016; Lavergne et al., 2016; Schwegmann et al., 2011; Sumata et al., 2014). However, the
Hudson Bay system is different from areas where drift accuracy has been studied. First, Hudson Bay has a smaller area to
shoreline ratio due to its smaller size compared to the rest of the Arctic Ocean (excluding the Canadian Archipelago), which
confounds satellite and wind-based drift estimation (Thorndike and Colony, 1982; Tschudi et al., 2020). Satellite-based
tracking relies on a feature-matching algorithm and cannot resolve velocities near the shore (Heil et al., 2001; Meier et al.,
2000; Tschudi et al., 2020). While currently NCEP wind is weighted half as much as buoy or satellite data, Tschudi et al.
(2020) noted that wind-based estimates are comparable to satellite estimates and may need to be given a higher weight.
Although giving wind estimate higher weight may improve drift estimates in Hudson Bay, it may still result in speed
underestimation. Wind-based drift estimates assume a 20° relationship with direction and a 1 % relationship with speed,
although this speed relationship may actually be higher (up to 3 %; Bai et al., 2015; Rabinovich et al., 2007). The effect of
wind on drift also varies depending on proximity to shore and the orientation of wind relative to the shoreline. Near the coast,
internal ice stress/forces can exceed those of wind and currents, with the effects extending up to 400 km (Fissel and Tang,
1991; Overland and Pease, 1988; Rabinovich et al., 2007; Thorndike and Colony, 1982). More complex regression-based
models that account for proximity and orientation of shorelines have been shown to improve wind-based drift estimates

(Rabinovich et al., 2007). Second, the bay is a seasonal system, completely melting in summer and reaching nearly 100 % cover in winter (Danielson, 1971; Saucier et al., 2004; Stewart and Barber, 2010). Consequently, sea ice in Hudson Bay lacks multi-year ice, and the ice is younger and generally thinner, with extensive periods of low concentration, factors which both decrease accuracy of modelled ice drift (Durner et al., 2017; Mahoney et al., 2019; Sumata et al., 2014). At low ice concentrations, satellites sensors are more likely not to detect sea ice (Castro De La Guardia et al., 2017; Tivy et al., 2011). The formation of new sea ice during freeze-up and the melt ponds that form during break-up both confound estimation of drift (Meier et al., 2000; Tschudi et al., 2020; Willmes et al., 2009). Third, there are no IABP buoys in Hudson Bay to contribute data to the NSIDC drift model, another factor associated with poorer model performance (Mahoney et al., 2019; Tschudi et al., 2020). Earlier versions of NSIDC drift products (see Tschudi et al., 2016) effectively limited the influence of buoys to ~ 350 km, which introduced artefacts around buoy locations (Szanyi et al., 2016). Changes to the algorithm in version 4 of NSIDC drift eliminated the artefacts and increased accuracy within the Arctic Ocean (Tschudi et al., 2020), however, these changes would not have improved drift estimates in regions without buoy data, including Hudson Bay. Last, the EASE-Grid projection is polar azimuthal and induces meridional compression and zonal stretching, which further biases drift estimation. The effect of this distortion is that north-south (east-west) drift is more likely to be underestimated (overestimated) and direction estimates will be biased toward the east-west axis. This bias is amplified as you approach the equatorial limits of dataset and is particularly important if groundspeed is required. Hudson Bay is the furthest body of water from the poles where NSIDC drift is estimated and would therefore experience the greatest bias due to projection. In summary, our observed speed underestimation may be explained by the challenging topography of Hudson Bay for satellite and wind-based drift estimates, underestimation of wind's impact on ice motion, small weight given to the wind input data, lack of buoy data, and projection biases.

A common limitation of these types of studies is the reliance on interpolation. Bilinear, or inverse distance weighted, interpolation yields estimates that tend towards the mean and precludes obtaining outermost estimates (Schwegmann et al., 2011). In addition, interpolation within skewed distributions is likely to yield spurious estimates. For example, in right-skewed datasets (e.g., zero-bound drift speed), outliers are more likely greater than the mean and inverse-distance averaging is more likely to be an overestimate. Nevertheless, there is no reason to believe these biases would be greater than those of other sea ice drift validation studies that used linear interpolation to match satellite with in situ based estimates (Lavergne et al., 2016; Schwegmann et al., 2011).

The drift biases we report are limited by availability of telemetry collar data, and we cannot definitively extrapolate our accuracy estimates beyond this spatiotemporal extent. Nevertheless, many of these biases have been reported in research of NSIDC and other satellite-based sea ice drift estimates (Heil et al., 2001; Karlsson, 2016; Lavergne et al., 2016; Linow et al., 2015; Rozman et al., 2011; Schwegmann et al., 2011; Sumata et al., 2014, 2015b, 2015a; Szanyi et al., 2016). Areas with similar characteristics to Hudson Bay may show similar biases in the estimated speed and direction of drift. This includes other seasonal systems (e.g., Baffin Bay), and those with slower drift (e.g., Kara and Laptev Seas) or without IABP buoys (see IABP, 2020 and Rampal et al., 2009 for coverage). Further, we observed the relative degree of bias increases with speed. If such

scaling in bias exists in other areas, then the magnitude of underestimation may be greater in areas with faster speeds (e.g., Chukchi Sea).

Assuming the overall NSIDC drift accuracy is consistent over time, these data are likely well-suited for addressing questions where the *relative* speed or direction are sufficient, for example longitudinal analyses such as climate-induced changes in drift speed (e.g., Kwok et al., 2013; Klappstein et al., 2020). Still, large error may obscure underlying trends. We

suggest cautious application of the NSIDC drift data where the *absolute* speed or direction is critical. For example, calculation of animal energetics (e.g., Durner et al., 2017; Klappstein et al., 2020), home ranges (e.g., Auger-Méthé et al., 2016a), voluntary movement (e.g., Togunov et al., 2017, 2018), and predicting/retrodicting distribution of drifting matter (Kohlbach et al., 2017; Peeken et al., 2018; Thorpe et al., 2007; Tschudi et al., 2010). The degree of error/bias that is permissible is research specific. Generally, to be able to correctly account for measurement error, it has to be smaller than the natural stochasticity of the system

being studied (Auger-Méthé et al., 2016b). Particular attention to error/bias should be given in regions without IABP buoy data or where bias is unquantified.

## 5 Conclusions

This study provides the first error estimates of any sea ice drift model in Hudson Bay. Using passively drifting telemetry collars, we quantified the accuracy and precision of Polar Pathfinder Daily 25 km EASE-Grid Sea Ice Motion Vectors (Version

4). Both u and v components of NSIDC drift along with the resultant speed tended to systematically underestimate true drift speed, a pattern exacerbated at higher speeds. The direction showed no systematic bias, however directional precision decreased at lower speeds. We suggest that any research requiring absolute values for drift speed/direction should account for error/bias of drift in the study design and/or test the sensitivity of the results to these biases (Cressie et al., 2009).

Although our collar GPS data were collected with the intent of studying polar bear ecology, we believe it and other

forms of animal-borne telemetry can be of great utility in advancing environmental modelling. For example, polar bear telemetry has been used to validate sea ice drift in the Beaufort and Chukchi Seas (Durner et al., 2017; Tschudi et al., 2010) and assess accuracy of sea ice concentration data (Castro De La Guardia et al., 2017), and seabird tracking has been used to estimate ocean currents and wind velocities (Goto et al., 2017; Yoda et al., 2014; Yonehara et al., 2016). In addition to being useful for model validation, these types of data can be incorporated into environmental models as additional data streams,

providing insight into areas that are more difficult to measure (Harcourt et al., 2019; Miyazawa et al., 2015). To help improve modelled drift data, we have made the position data of our drifting collars public https://doi.org/10.7939/dvn/kuiz7g. The data can also be used to identify error/bias associated with different locations, periods, or environmental conditions (e.g., ice thickness, ice concentration, and cloud cover) in which models can be improved (e.g., Miyazawa et al., 2015). Our study provides evidence of modelled ice drift bias in Hudson Bay, where lack of Arctic buoys makes this type of study difficult.

Ultimately, these findings (in combination with our public data set and that of other drifting tag data; Durner et al., 2017;

Øigård et al., 2010; Vacquie-Garcia et al., 2017) can be a good resource for quantifying and validating the accuracy of other and/or future ice drift products.

**Data availability**

The Polar Pathfinder Daily 25 km EASE-Grid Sea Ice Motion Vectors (Version 4) dataset is available at
(https://nsidc.org/data/nsidc-0116/versions/4, last access: 5 April 2020). The location data of the passively drifting collars is available at (https://doi.org/10.7939/dvn/kuiz7g, last access: 5 April 2020)

**Author contributions**

RT identified the drifting collars. RT and NK designed the study and conducted the analyses with contributions from MAM and AD. NL and AD conducted field work with assistance from RT and NK. RT prepared the manuscript with contributions
from all authors.

**Competing interests**

The authors declare that they have no conflict of interest.

**Acknowledgments**

Financial and logistical support of this study was provided by Canadian Association of Zoos and Aquariums, the Canadian
Research Chairs program, the Churchill Northern Studies Centre, Canadian Wildlife Federation, Care for the Wild International, Earth Rangers Foundation, Environment and Climate Change Canada, Hauser Bears, the Isdell Family Foundation, Kansas City Zoo, Manitoba Sustainable Development, Natural Sciences and Engineering Research Council of Canada, Parks Canada Agency, Pittsburgh Zoo Conservation Fund, Polar Bears International, Quark Expeditions, Schad Foundation, Sigmund Soudack & Associates Inc., Wildlife Media Inc., and World Wildlife Fund Canada.

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

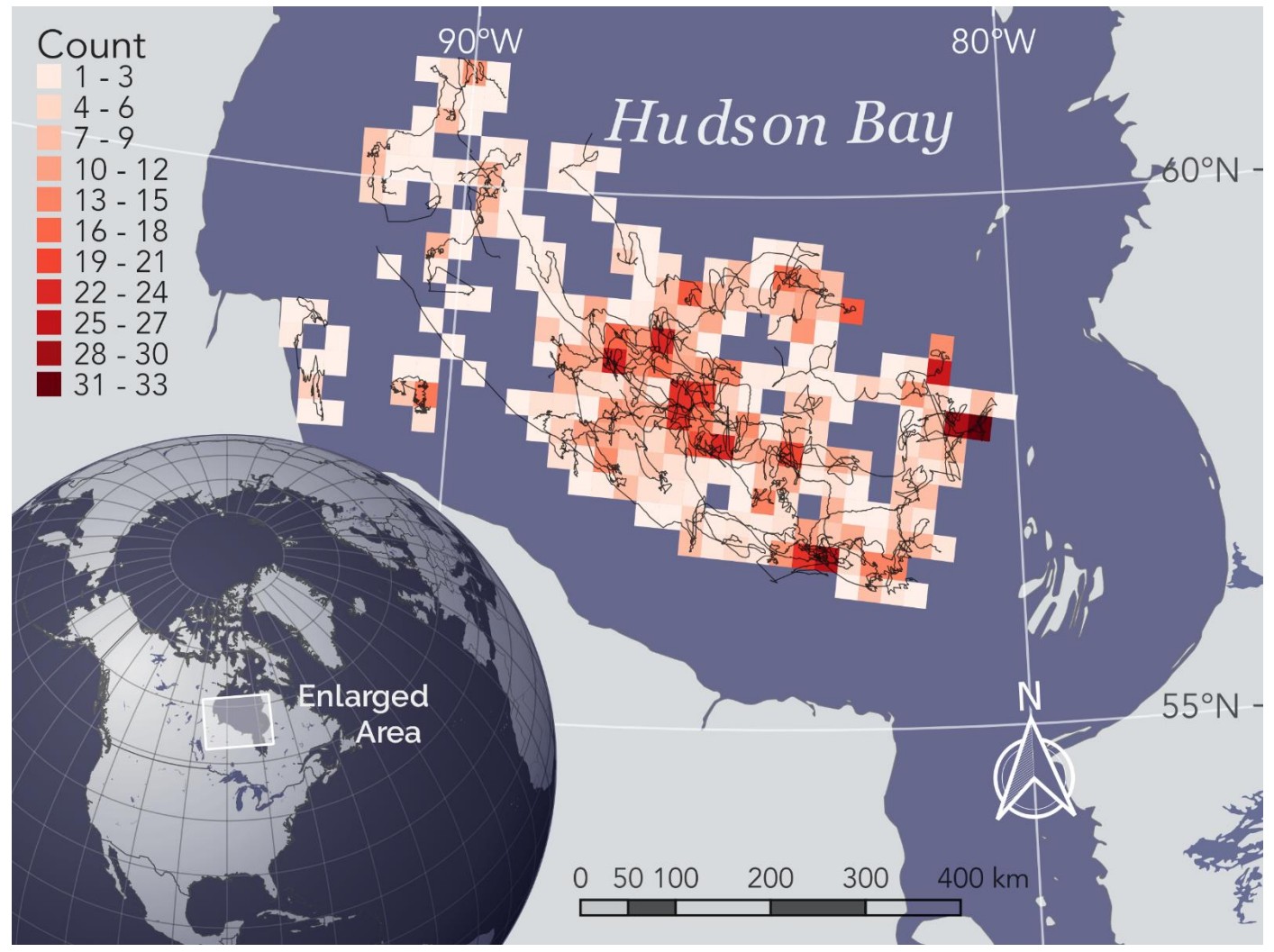

**Figure 1. Hudson Bay study area (enlarged), tracks of dropped collars (black lines), and count of drift vectors (shaded cells, projected in 25 km EASE-grid North, EPSG: 3408). World borders dataset obtained from Sandvik (2009).**


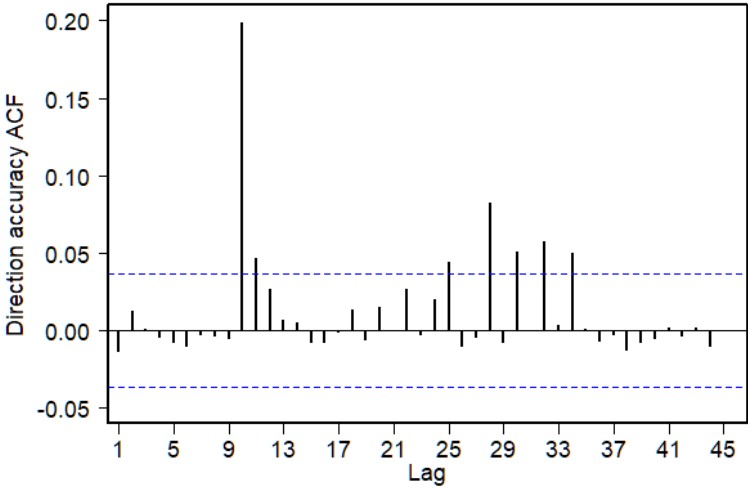

**Figure 2. Auto-correlation function (ACF) for NSIDC linearized direction accuracy, $tan(|Direction_{NSIDC} - Direction_{collar}|/2)$. Blue lines correspond to the 95% CI limits that represent significant autocorrelation.**

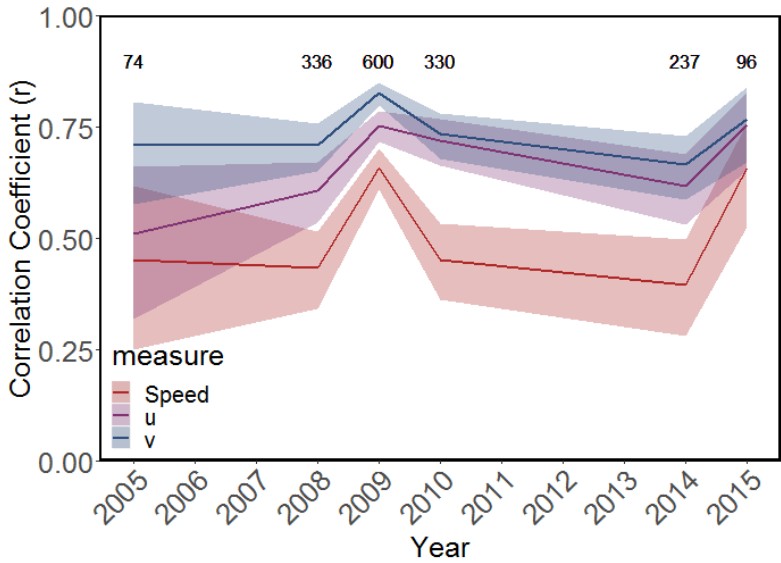

**Figure 3. Interannual variation in correlation coefficients (r) between NSIDC drift and collar drift speed (red line), u component (purple line), and v component (blue line). Shaded areas represent the 95% CI of the correlation coefficient. Numbers at the top represent the number of drift vectors compared in each year. 2013 excluded due to insufficient data n = 4.**

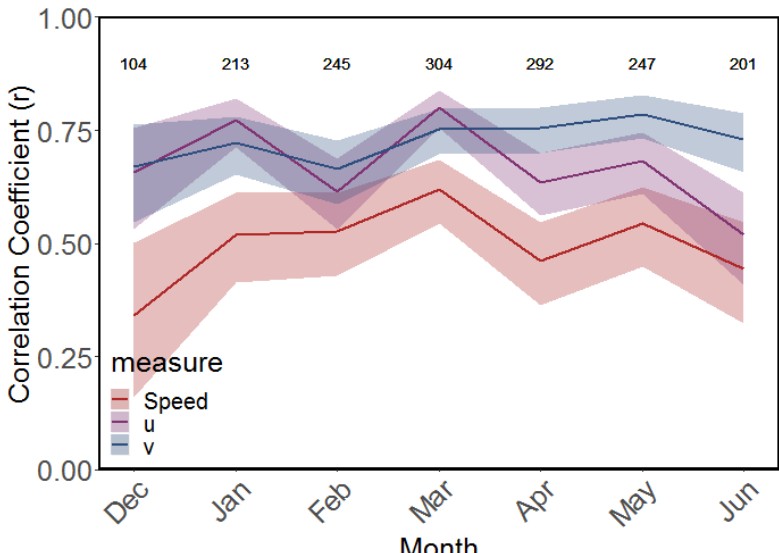

**Figure 4. Intra-annual variation in correlation coefficients (r) between NSIDC drift and collar drift speed (red line), u component (purple line), and v component (blue line). Shaded areas represent the 95% CI of the correlation coefficient. Numbers at the top represent the number of drift vectors compared in each month. July excluded due to insufficient data n = 71.**

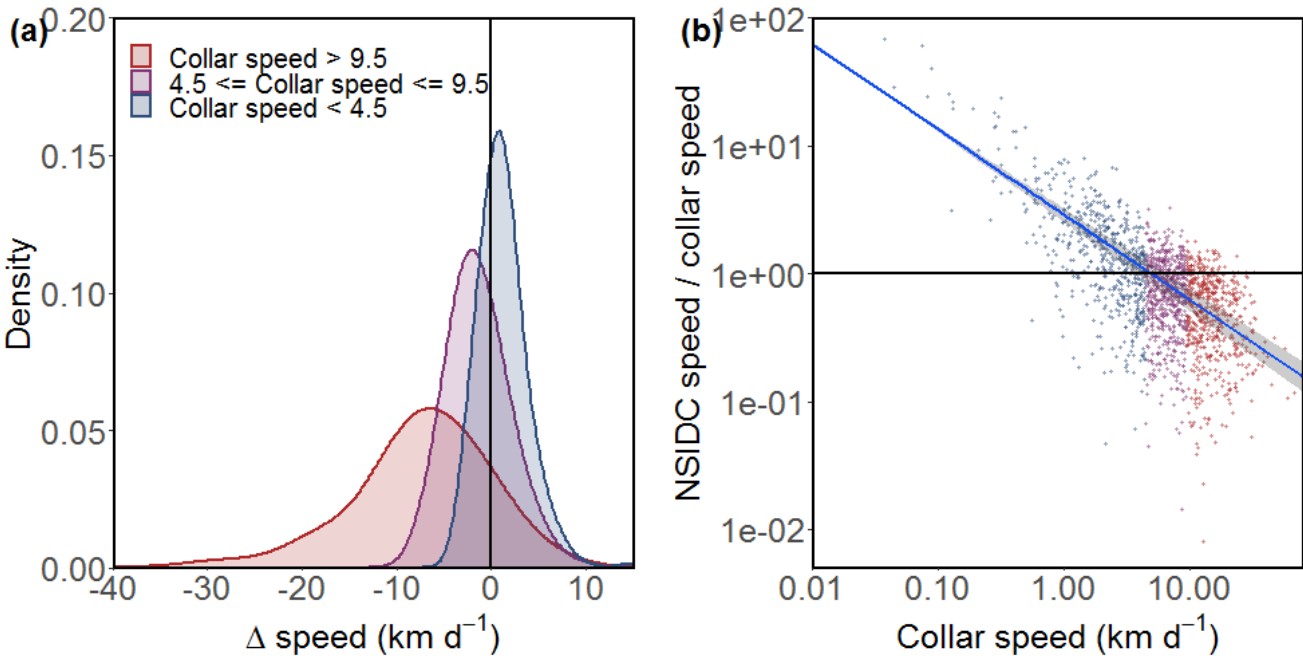

**Figure 5. Accuracy of NSIDC drift speed represented by (a) histogram and density plot of the absolute accuracy ($Speed_{NSIDC} - Speed_{collar}$) and (b) GLMM$_{PQL}$ of relative accuracy ($Speed_{NSIDC}/Speed_{collar}$) as a function log-transformed collar speed (presented on log-log scale; blue line is the GLM$_{PQL}$ prediction of the mean with shaded 95 % CI). In both A and B, data points are separated into three groups (red, purple, and blue) based on collar speed to convey speed-specific variability in accuracy. Black lines represent 1:1 unanimity between NSIDC and collar drift speeds.**

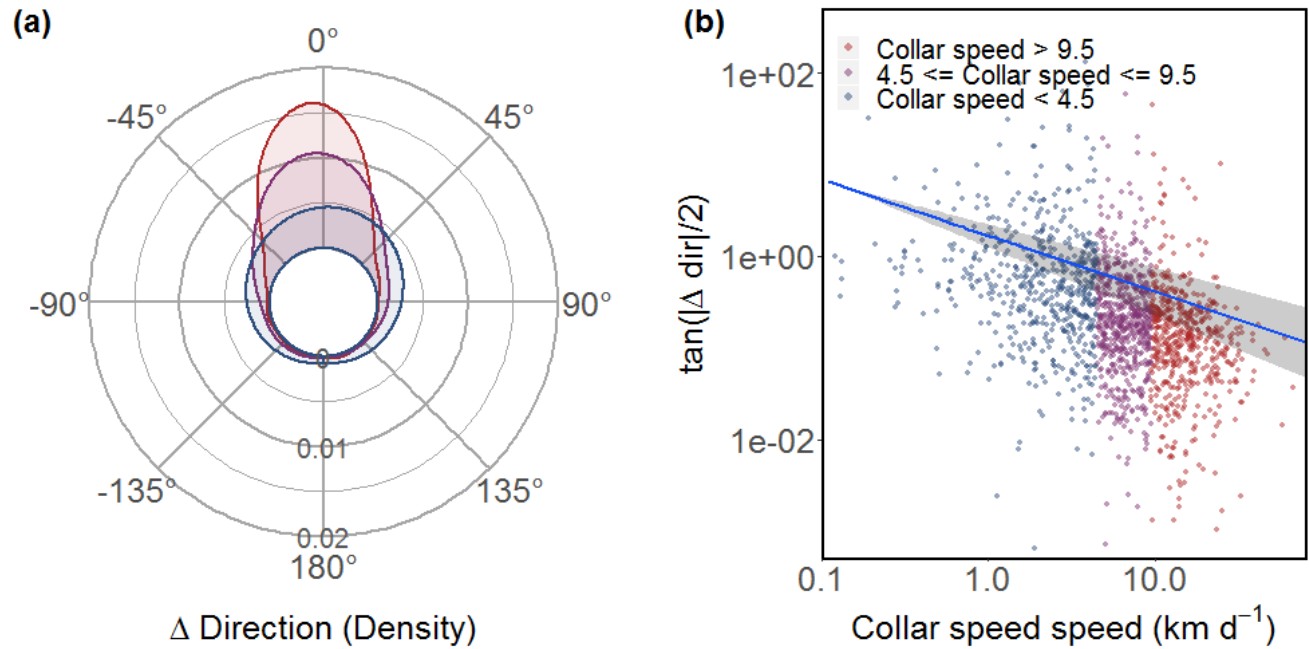


**Figure 6. Accuracy of NSIDC drift direction represented by (a) circular histogram and density plot of the absolute accuracy ($Direction_{NSIDC} - Direction_{collar}$) and (b) GLMM$_{PQL}$ of relative accuracy ($tan(|Direction_{NSIDC} - Direction_{collar}|/2)$) as a function of log-transformed collar speed (presented on a log-log scale, with a zero value representing 1:1 unanimity); blue line in represents the GLMM$_{PQL}$ prediction of the mean with the shaded area representing the 95 % CI. Data points are separated into**
**three groups (red, purple, and blue) based on collar speed to convey speed-specific variability in accuracy.**

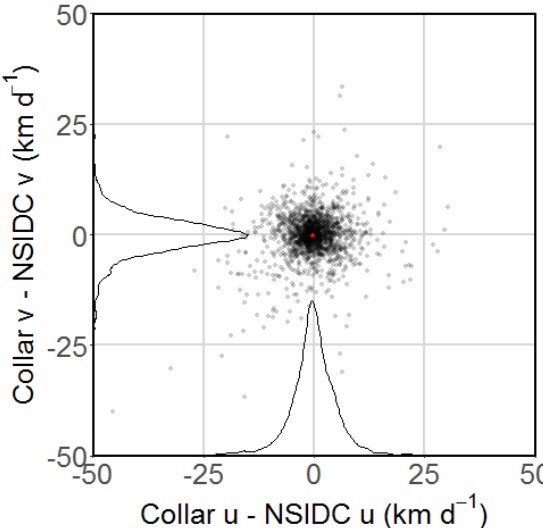

**Figure 7. Difference between collar drift and NSIDC drift for the u (x-axis) and v (y-axis) components. Curves represent density of differences and the red dot represents the mean difference of u and v components.**

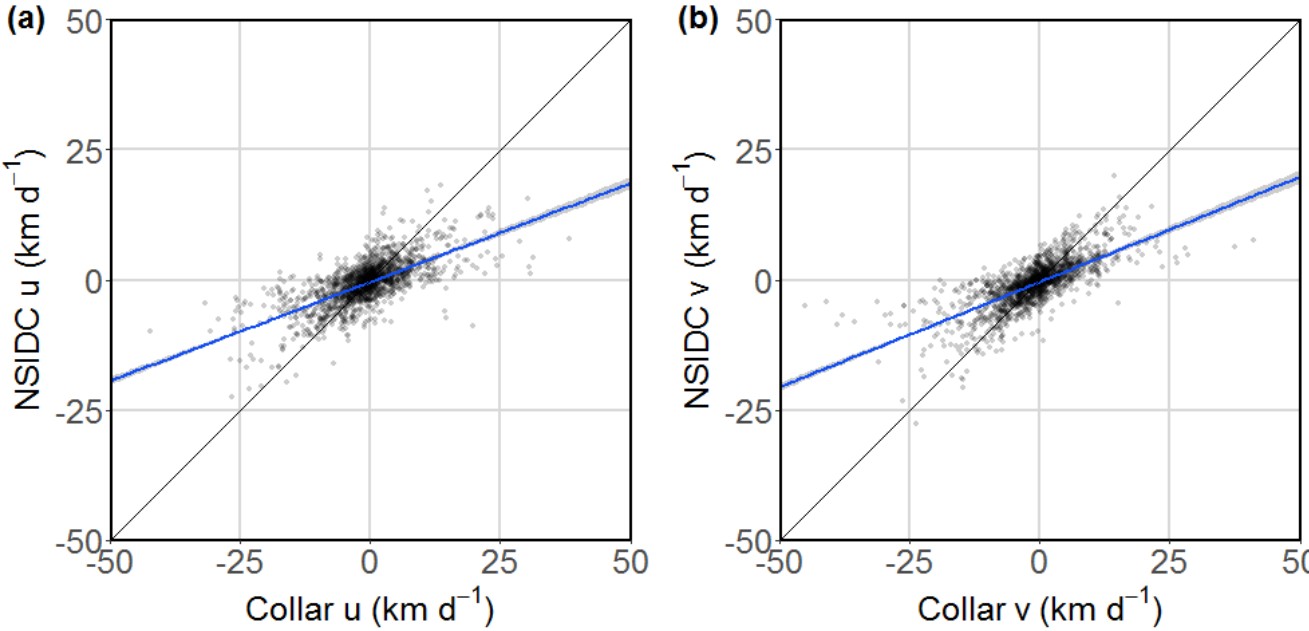


**Figure 8. GLMM$_{PQL}$ (family: Gaussian) regression of the u (a) and v (b) components of NSIDC drift vector versus collar drift. Black lines represent a 1:1 relationship between NSIDC and collar drift components; the blue lines represent the lines of best fit with the shaded areas representing 95 % CI of the mean.**

## Appendix A. Drifting collar identification

Collars deployed since 2011 were equipped with 'activity sensors' that are triggered following an extended period of inactivity. These collars were considered passively drifting if the activity sensor turned on and stayed until the end of transmissions. For collars deployed before 2011 had to be identified manually in two stages.

First, GPS location data were annotated with sea ice motion vectors from NSIDC's Polar Pathfinder Daily 25 km EASE-Grid Sea Ice Motion Vectors, Version 3 (http://nsidc.org/data/nsidc-0116). Daily drift estimates were spatiotemporally interpolated to match the location and time of GPS fixes - it was assumed that the ice motion data reflected average drift at noon of each day. For all 4 h GPS fixes, voluntary bear movement was estimated by subtracting the component of ice drift from the GPS displacement. This estimate of voluntary movement was plotted against time for each collar. Collars were suspected to be drifting if there was a sudden and sustained drop in movement speed (e.g., Fig. A1 versus Fig. A2). To confirm that the collars are indeed drifting, the displacement of these suspect collars had to be confirmed to reflect the actual sea ice drift.

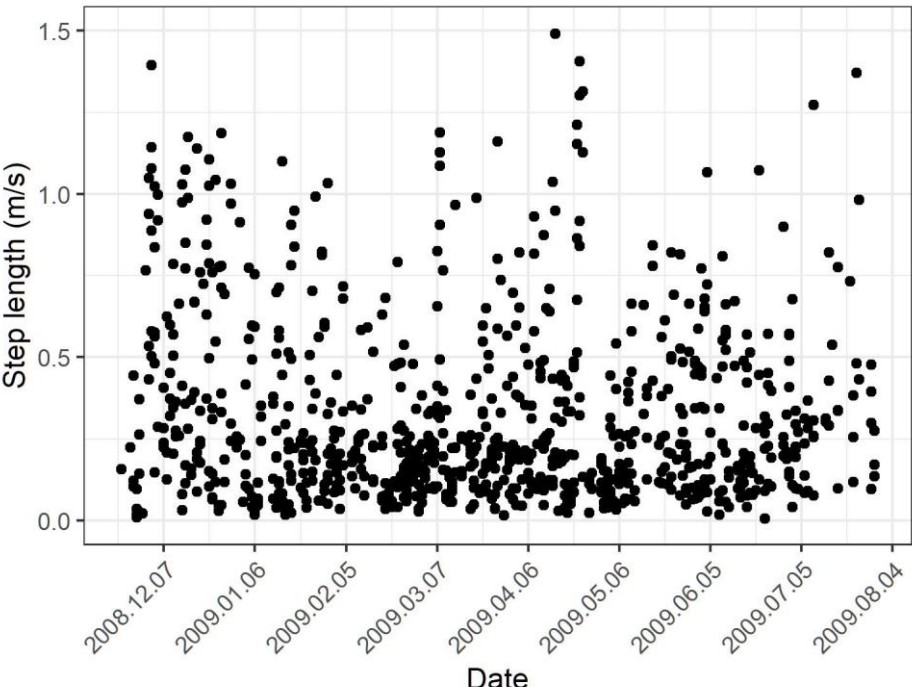

**Fig. A1. Example of estimated voluntary movement (step length in m s⁻¹) over time of a collar that is on a living bear.**

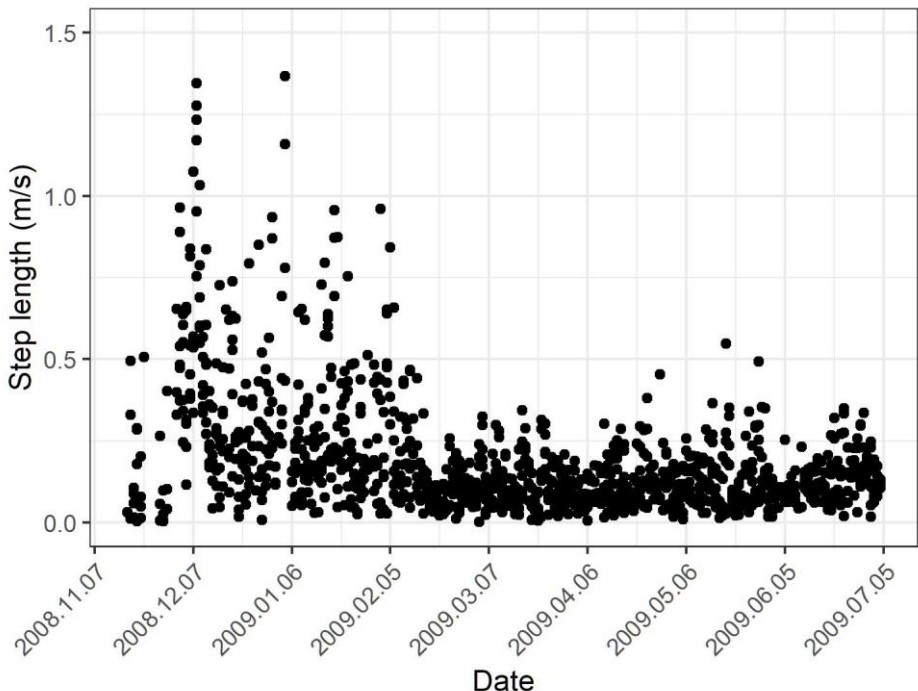

**Fig. A2. Example of estimated voluntary movement (step length in m s⁻¹) over time of a suspect passive collar.**

Actual sea ice drift was derived from NASA's Earth Observing System Data and Information System (EOSDIS) satellite imagery (https://earthdata.nasa.gov/about). First, the projection and scale of EOSDIS and collar locations had to be matched. EOSDIS Worldview web interface (https://worldview.earthdata.nasa.gov/) projection was set to "Arctic" (WGS 84 / NSIDC Sea Ice Polar Stereographic North projection; EPSG: 3413), rotated -69°, and was maximally zoomed in. Collar locations were plotted in QGIS Version 2.16.3, the projection was matched to EOSDIS (EPSG: 3413) and scaled in QGIS to 1:480,000 (though the realized scale was ~ 1:1,330,000 on the 13.3-inch computer at a 2560 × 1600 resolution).

Next, sea ice drift was estimated at a subset of locations for each suspected drifting collar using the following procedure. First, the view in QGIS was centred on GPS locations of a probably drifting collar where ice drift would be approximated and the view in EOSDIS Worldview was matched. Second, we identified periods where the satellite imagery was relatively unobscured by clouds for at least two days and visually tracking ice floes would be possible. Third, a collar location representing the first location of a displacement vector (hereafter, first-day collar location) was marked using the screen annotation software AnnotatePro (http://www.annotatepro.com/). Fourth, we identified unique sea ice features that could be tracked over both days. Unique ice features were mainly distinctive edges and corners of ice floes and fractures. Fifth, using AnnotatePro, we marked where an ice floe was on the day of the collar location (hereafter, first-day ice location) and another point was marked where that same ice floe was on the following day (hereafter, second-day ice location). Sixth, both marks were selected using selection tool in AnnotatePro and moved such that the first-day ice location overlapped the first-day collar location. The second-day ice

location represented where the collar would be located on the following day had the bear not moved. If the collar location was on an identified ice floe, only that floe was tracked. If the collar location was not on an identified floe, up to five additional floes around the collar location were identified and marked to attain an approximation of drift at the collar location. Seventh, the distance between second-day ice location and the second-day collar location was calculated using the 'measure line' tool in QGIS. If several ice floes were marked and tracked, then the distance was measured from the second-day collar location to the approximate centre of all the second-day ice floe locations. At the operating scale being used, sea ice drift was relatively uniform and there was very high consistency in drift among ice floes.

Collars were assumed to be passively drifting collars if the mean of at least four consecutive distance estimates (hereafter, distance estimate) was < 2 km (hereafter, distance threshold). At the maximum resolution permitted in EOSDIS, the 2 km distance threshold corresponded to ~ 1.5 mm on screen. If the distance estimate was greater than the distance threshold, the collar was assumed to be on a live bear and not a drifting collar.

The EOSDIS imagery used was taken during daylight hours, so sea ice drift was estimated (as much as possible) for collar locations at 17:00 and 21:00 UTC, generally corresponding to midday in Hudson Bay. For each suspect drifting collar, sea ice drift was first estimated for the last days of collar locations; if the distance estimate was greater than the distance threshold (i.e. indicating a live bear), all prior locations must also have been on a live bear. If the distance estimates were less than the distance threshold the collar was assumed to be drifting, then drift was estimated iteratively ~ 30 d into the past until the distance estimate indicated a live bear. Next, from the last date assumed to be drifting, sea ice drift was estimated iteratively ~ 7 d into the past until the mean distance estimate indicated a live bear. Finally, from the last drifting collar date, I examined prior days sequentially until the distance estimate indicated a live bear. The following day was determined to be the date when the collar either dropped off the bear or the bear died.

For certain days, ice drift estimation was either very poor or not possible. Confounding factors included: heavy cloud cover, blurry satellite imagery, small floes that were indistinguishable and not trackable (particularly common during freeze-up and break-up), consolidated ice with no trackable features, or days with extreme fracturing of ice floes beyond recognition. For these periods, certain modifications to the described protocols were permitted. For example, if cloud-free days were separated by up to two clouded days and sea ice drift could be estimated across that period, this was permitted. If many of the drift estimates were poor, researcher discretion was permitted to increase the drifting collar threshold from 2 km. During periods with extensively poor ice drift estimation, if four sequential drift estimates spanned beyond a week, it was permitted to average fewer than four estimates.

## Appendix B. Drifting collar validation

To lend additional support that manually identified collars were indeed not on active bears, we compared metrics of speed, direction, and u/v component accuracy calculated for manually identified collars, activity sensor identified collars, and active collars. First, we subset the active collars to a 24 h resolution by filtering only fixes obtained at 13:00 UTC. Second, calculated the displacement vectors (speed, direction, and u/v components; calculated in the EASE-Grid North projection, EPSG: 3408) between successive days, then we filtered any vectors representing displacement over > 24 h. Third, we subset the active collar vector data to the same number of locations as the drifting collars (n = 1677) and only in the years (2005, 2008-2010, and 2013-2015) and months (December-June). These data were then compared to drifting collars identified manually and using the activity sensor.

The metrics of comparison were speed accuracy ($Speed_{NSIDC} - Speed_{collar}$; Fig. B a), direction accuracy ($Direction_{NSIDC} - Direction_{collar}$; Fig. B b). We also tested the correlation in speed, direction, u component, and v component between NSIDC drift estimates and collar displacement vectors (Fig. B c and Fig. B). For speed, we calculated the Pearson's correlation coefficient (Fig. B c). For direction, we calculated the circular Pearson's correlation coefficient ($\pm$ 95% CI) using the 'cor.circular' function in the 'circular' package in R. We used bootstrapping with 1000 replicates to calculate 95% CI for this circular correlation (Fig. B c). As an additional metric of directional accuracy, we estimated the concentration parameter (kappa $\pm$ 95% CI) on the difference between NSIDC drift and collar displacement vectors (Fig. B c). Last, we fit a GLMMPQL (family: Gaussian) with the NSIDC drift u and v components as functions of u and v components of active, manually identified, and activity sensor identified collars (Fig. B).

There were no significant differences between manually identified drifting collars (n = 13) and collars identified using activity sensor (n = 7) in accuracy metrics of speed, directional, or u and v components. However, both manually and activity sensor identified collars were consistently significantly different from collars on active bears with regard to the same accuracy metrics (Fig. B and Fig. B). All results exhibit a significantly weaker relationship between NSIDC drift and displacement of active collars compared to either passively drifting collars.

The motion for six manually identified collars is depicted in the supplement video (http://doi.org/10.5446/45186). This video depicts the large degree of concurrence of drift vectors across large spatial extent, and further lends evidence that the manually identified collars are in fact passively drifting.

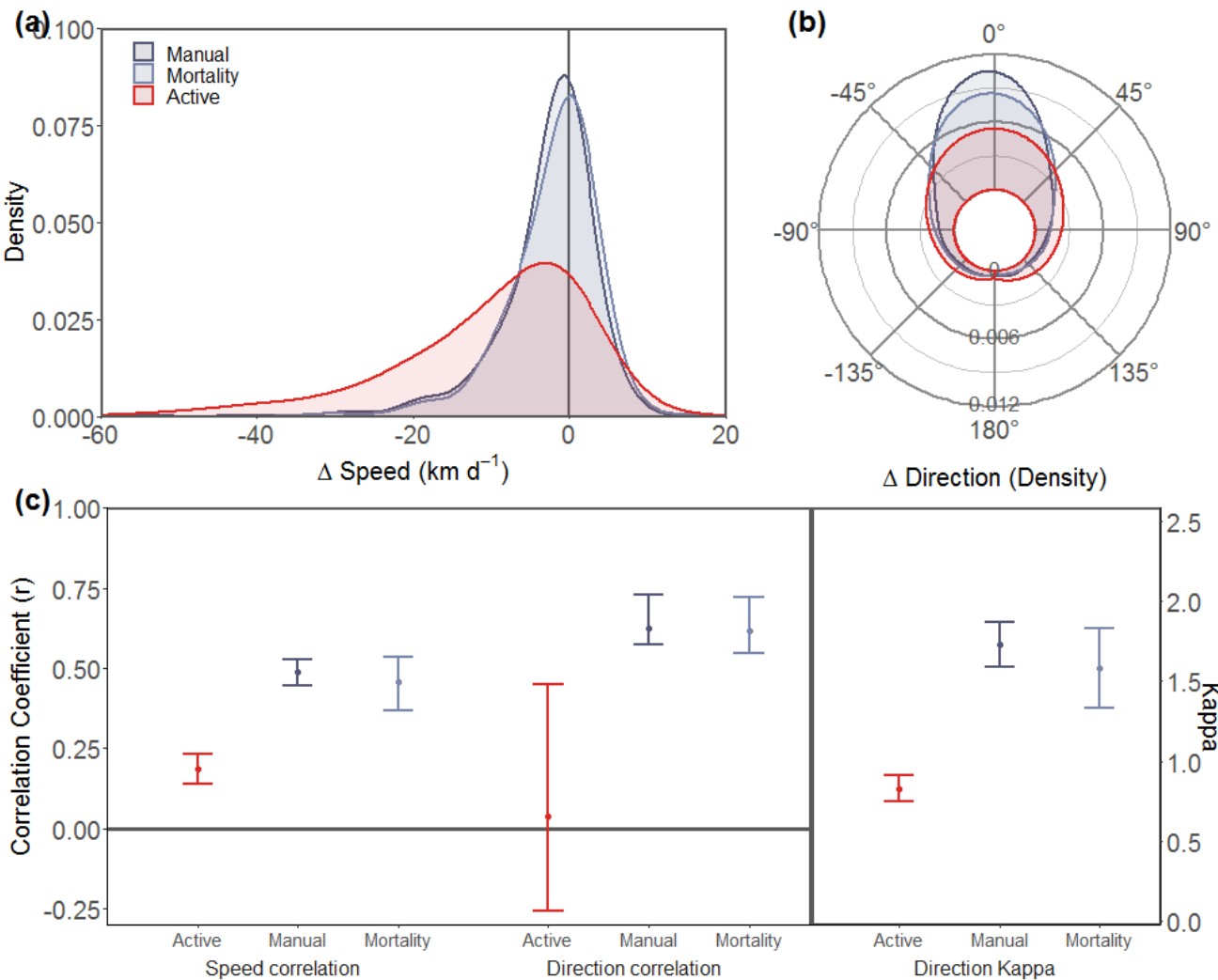

**Fig. B1. Comparison of speed and direction metrics of collars believed to be on active bears (red), manually identified drifting collars (dark blue), and 'activity sensor' identified drifting collars (light blue). Metrics presented are density plot of the difference in speed, ($Speed_{NSIDC} - Speed_{collar}$; A), density plot of difference in direction ($Direction_{NSIDC} - Direction_{collar}$; B), Pearson's correlation coefficients of speed ($Speed_{NSIDC} \sim Speed_{collar}$; C, left) and direction ($Direction_{NSIDC} \sim Direction_{collar}$; C, middle), and estimated of angular concentration (kappa) in the difference in direction (C, right). Error bars in C represent 95% CI of the mean.**

595

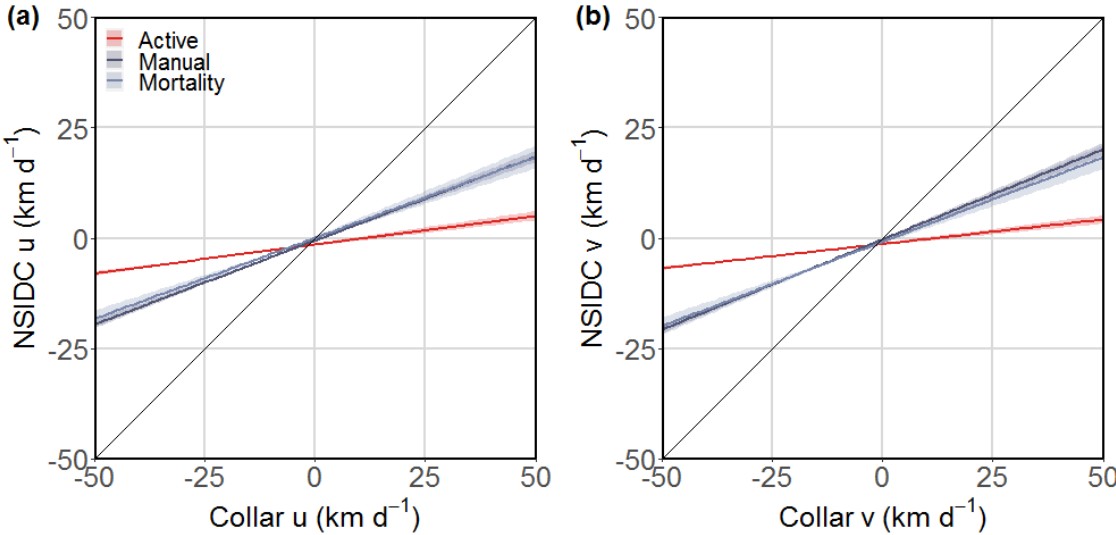

**Fig. B2. GLMM$_{PQL}$ regression of the u (A) and v (B) components of NSIDC drift vector versus collar drift among collars believed to be on active bears (red), manually identified drifting collars (dark blue), and 'activity sensor' identified drifting collars (light blue). Black lines represent a 1:1 relationship between NSIDC and collar drift components; Shaded areas representing 95% CI of the mean.**

600