# Peer review of "Opportunistic evaluation of modelled sea ice drift using passively drifting telemetry collars in Hudson Bay, Canada"

_The Cryosphere, 2020_

## Referee Comment (RC1) · Anonymous Referee #1 · 2 Mar 2020

Togunov et al,. 2020 In general: This paper evaluates the bias/accuracy of the NSIDC sea ice motion product via comparisons with drifting GPS collars in Hudson Bay. The paper is well-written and referenced, and uses a robust set of statistics to evaluate the product. The discussion details the characteristics of Hudson Bay, which likely affect the performance of the motion product in this region. However, the authors note that their findings are not divergent from other evaluations of this product in the Arctic Ocean. I recommend publication of this paper, with minor edits as indicated below.

Line # Comment 1 Needs indent 64 NSIDC model? 66 I would argue that satellite-based estimates are not validation data, rather, inter-comparison data, since they are

[Figure]

estimates with their own set of biases. The in situ (2) and (3) obs are suitable for validation. 72 What studies? Reference(s)? 74 I'm not certain that the more recently-deployed IABP buoys have these types of errors. The ref was published in 2013, and there have been improvements to these buoys since then. 166 Interesting. NSIDC ice motion does incorporate a turning of the sea ice left, so here it appears the angle of turn may be too high in this region 205 Okay, and the turning angle is discussed here 210 melt ponds 405 I think you used Version 4, so need to update this reference

---

## Referee Comment (RC2) · Anonymous Referee #2 · 5 Apr 2020

The manuscript aims to assess the accuracy in the sea ice drift products provided by NSIDC, by comparing them to drift collars. The manuscript is well written and clear.

The area investigated is relatively limited. How are the drift speeds compared to other regions, i.e. would the underestimate also extend to other regions and what would the effect then be on the Pan-Arctic scale?

There are some minor comments that needs to be addressed before publication. Row 44. What is meant with ecology? Ecological studies? Row 87. 4h. Row 87. Are -> have a Row 99. The -> the Row 99. Replace – with for Row 100. AMSRE -> AMSR-E Row 156. Please explain what: intercept, CI, df, t and p stands for Row 181. What kind

of biases were identified? Row 199. Remove "but see"

Does TC allow for paper in review to be included in the reference list? If not please remove Klappstein et al. in review from the reference list.

Row 257-258. Please provide the link to the data here. Row 266. Please provide the link to the data here. Row 326. What type of reference is this? PhD thesis? Report?

---

## Author Comment (AC2) · 10 Apr 2020

$>>$ **We thank the reviewer for their quick response and valuable feedback.**
The manuscript aims to assess the accuracy in the sea ice drift products provided by
NSIDC, by comparing them to drift collars. The manuscript is well written and clear.

$>>$ **Thank you for this positive comment.**

The area investigated is relatively limited. How are the drift speeds compared to other
regions, i.e. would the underestimate also extend to other regions and what would the

effect then be on the Pan-Arctic scale?

$>>$ **We agree, the area investigated is relatively limited. As mentioned in our previous version "The drift biases we report are limited by availability of telemetry collar data, and we cannot definitively extrapolate our accuracy estimates beyond this spatiotemporal extent." (lines 219-220). Nevertheless, to expand on the generality of the results while accounting for this limitation, we have added the following statement regarding our observed underestimates extending to other regions:**
**"However, areas with similar characteristics may show similar biases in the estimated speed and direction of drift. This includes other seasonal systems (e.g., Baffin Bay), and those with slower drift (e.g., Kara and Laptev Seas) or without IABP buoys (see IABP, 2020 and Rampal et al., 2009 for coverage). Further, we observed the relative degree of bias increases with speed. If such scaling in bias exists in other areas, then the magnitude of underestimation may be greater in areas with faster speeds (e.g., Chukchi Sea)."**
**References:**
**IABP: International Arctic Buoy Program - Animated Buoy Movies, Univ. Washingt. [online] Available from: http://iabp.apl.washington.edu/data_movie.html (Accessed 9 April 2020), 2020.**
**Rampal, P., Weiss, J. and Marsan, D.: Positive trend in the mean speed and deformation rate of Arctic sea ice, 1979-2007, J. Geophys. Res. Ocean., 114(5), C005066, doi:10.1029/2008JC005066, 2009.**

There are some minor comments that needs to be addressed before publication.

$>>$ **No response needed.**

Row 44. What is meant with ecology? Ecological studies?

[Figure]

>> **Yes - now reads "ecological research."**

Row 87. 4h.

>> **The Cryosphere formatting guidelines require spaces between number and unit**

Row 87. Are $->$ have a; Row 99. The $->$ the; Row 99. Replace – with for; Row 100. AMSRE $->$ AMSR-E

>> **Changes made.**

Row 156. Please explain what: intercept, CI, df, t and p stands for

>> **Our methods explain that "we used an intercept-only [$GLMM_{PQL}$] ... wherein a significant intercept represents a significant difference between the model and the collar speeds." We have defined 'CI' and 'df' and clarified 't' and 'p' throughout the manuscript. Parenthetical now reads: "($GLMM_{PQL}$: intercept $\pm$ 95 % confidence interval (CI) = -3.0 $\pm$ 1.2 km d$^{-1}$, degrees of freedom (df) = 1657, t-value = -4.8, p-value $>$ 0.0001; Figure 5 )"**

Row 181. What kind of biases were identified?

>> **We have clarified the biases identified in the first discussion sentence, which now reads: "Using drifting collars as reference data for validation, we identified biases in the estimated speed and direction of NSIDC modelled sea ice drift model."**

Row 199. Remove "but see"

[Figure]

>> **Change made.**

Does TC allow for paper in review to be included in the reference list? If not please
remove Klappstein et al. in review from the reference list.

>> **This paper is now in press and its reference updated.**

Row 257-258. Please provide the link to the data here.

>> **The data is available at https://doi.org/10.7939/DVN/KUIZ7G. This link has
been added in text.**

Row 266. Please provide the link to the data here.

>> **Change made.**

Row 326. What type of reference is this? PhD thesis? Report?

>> **This is a BSc thesis. The citation now includes "BSc Thesis, Department of
Physics Lund University"**

---

## Author Response (AR1)

**Author's Response - "Opportunistic evaluation of modelled sea ice drift using passively drifting telemetry collars in Hudson Bay, Canada" by Ron R. Togunov et al.**

Response to Editor

We thank the reviewers and the editor for your quick response and valuable feedback. We have developed the discussion as suggested and clarified and corrected ambiguous sections. We have also added two citations to the conclusion that also note having passively drifting tag data (Øigård et al., 2010; Vacquie-Garcia et al., 2017).

Øigård, T. A., Haug, T., Nilssen, K. T. and Salberg, A. B.: Estimation of pup production of hooded and harp seals in the Greenland Sea in 2007: Reducing uncertainty using generalized additive models, J. Northwest Atl. Fish. Sci., 42, 103–123, doi:10.2960/J.v42.m642, 2010.
Vacquie-Garcia, J., Lydersen, C., Biuw, M., Haug, T., Fedak, M. A. and Kovacs, K. M.: Hooded seal *Cystophora cristata* foraging areas in the Northeast Atlantic Ocean-Investigated using three complementary methods, , doi:10.1371/journal.pone.0187889, 2017.

Comments to the Author:

In addition to addressing the reviewer comments in your revised manuscript, I have one important comment and a few minor comments below:

Note that the journal policy on supplemental material has recently been updated (see here https://www.the-cryosphere.net/for_authors/manuscript_preparation.html). In particular, the policy states "Normal size figures, tables, as well as technical or theoretical developments that do not need to be included in the main text should be included as appendices". Please move the supplemental material to an appendix.
We moved the supplementary material into the manuscript as two appendices.

Line 204 –wind-based estimates have a range; I think most are more like 2% of the wind speed, and the direction is often variable. You might also want to explain why this is relevant here.
We have added further discussion on wind and ice drift (see lines 209 – 218).
For clarity, we added a brief summary sentence at the end of this paragraph that reads (lines 235 – 238): "In summary, our observed speed underestimation may be explained by the challenging topography of Hudson Bay for satellite and wind-based drift estimates, underestimation of wind's impact on ice motion, small weight given to the wind input data, lack of buoy data, and projection biases."

I think NSIDC relies on winds only in summer, and the turning angle they use is 20 degrees.
We have corrected the turning angle to 20°. We have confirmed that winds are applied all year in the Northern Hemisphere through personal communication with the data producers.

Line 227-229 – Can you clarify if this bias actually exists? It is true that the grid will produce distortion. But, the NSIDC drift vectors are provided as actual u and v velocities. I thought these are computed from the actual physical positions, not from the EASE-grid grid positions. If it is the former, then the

projection is not relevant, if the latter, then I agree, there is some distortion (in which case no change is needed)

We have verified that azimuthal distortion of the EASE-grid projection is not accounted for through personal communication with the data producers. According to the online documentation, satellite data is first projected into EASE-Grid Lambert Azimuthal (EPSG: 3408), then using maximum cross correlation (MCC) of a 10 x 10 pixel window, the "location with the best correlation coefficient… is considered the ice displacement" (NSIDC 2020). The documentation defines u (v) as being "Along-x [y] component of the sea ice motion (not the eastward [northward] velocity)" and x and y to be in "Projected meters". All calculations are done in EASE-Grid Lambert Azimuthal, with no subsequent processing.

For clarity, we have moved this portion of the discussion into an earlier paragraph discussing potential sources of error in Hudson Bay.

NSIDC: Measuring Sea Ice Motion: https://nsidc.org/data/pm/nsidc0116-icemotion-smmr-ssmi, last access: 09 May 2020. 2020.

Figure 7 caption – first sentence is confusing. Perhaps "Difference between collar drift and NSIDC drift for the u (x-axis) and v (y-axis) components"?

Change made.

S1 third last sentence – do you mean "researcher discretion"?

This is correct - change made.

Response to Anonymous Referee #1

Togunov et al,. 2020 In general: This paper evaluates the bias/accuracy of the NSIDC sea ice motion product via comparisons with drifting GPS collars in Hudson Bay. The paper is well-written and referenced, and uses a robust set of statistics to evaluate the product. The discussion details the characteristics of Hudson Bay, which likely affect the performance of the motion product in this region. However, the authors note that their findings are not divergent from other evaluations of this product in the Arctic Ocean. I recommend publication of this paper, with minor edits as indicated below.

Line # Comment
1 Needs indent
Line 1 is the title and must be left-justified. If the reviewer was referring to the first line of abstract (14) or introduction (28); The Cryosphere format requires the first paragraph of each section to be left-justified.

64 NSIDC model?
Change made.

66 I would argue that satellite- based estimates are not validation data, rather, inter-comparison data, since they are estimates with their own set of biases. The in situ (2) and (3) obs are suitable for validation.
We agree with this. Sentence now reads (lines 66 – 69):
"There are two types of data that can be used to cross-validate ice drift: (1) other telemetry-based estimators including moored Doppler-based velocity measures and other high resolution satellites (e.g., Advanced Very High Resolution Radiometer (AVHRR) or Synthetic Aperture Radar (SAR)), and (2) in situ drifters, including buoys, ships, and manned stations."

72 What studies? Reference(s)?
Reference added. Sentence now reads (lines 73 – 75):
"Since there are few sources of in situ sea ice drift data, at least one study quantifying NSIDC drift accuracy used the same IABP data that are integrated into NSIDC model for validation, which may underestimate bias (e.g., Sumata et al., 2014)."

74 I'm not certain that the more recently deployed IABP buoys have these types of errors. The ref was published in 2013, and there have been improvements to these buoys since then.
This is true. Sentence now reads (lines 75 – 76):
"Further, IABP buoys have historically used ARGOS location estimates, which have spatial errors up to tens of kilometres and may be unsuitable for validation of drift during the periods/areas they were deployed"

166 Interesting. NSIDC ice motion does incorporate a turning of the sea ice left, so here it appears the angle of turn may be too high in this region
No comment necessary.

205 Okay, and the turning angle is discussed here
No comment necessary.

210 melt ponds
Change made.

405 I think you used Version 4, so need to update this reference
We do cite version 3 in the text so this citation will be kept, however, we have added an additional reference to the version 4 data set where applicable.

Response to RC2 – Anonymous Referee #2

205 Okay, and the turning angle is discussed here
No comment

The manuscript aims to assess the accuracy in the sea ice drift products provided by NSIDC, by comparing them to drift collars. The manuscript is well written and clear.
Thank you for this positive comment.

The area investigated is relatively limited. How are the drift speeds compared to other regions, i.e. would the underestimate also extend to other regions and what would the effect then be on the Pan-Arctic scale?
We agree, the area investigated is relatively limited. As mentioned in our previous version "The drift biases we report are limited by availability of telemetry collar data, and we cannot definitively extrapolate our accuracy estimates beyond this spatiotemporal extent." (now lines 252 – 253). Nevertheless, to expand on the generality of the results while accounting for this limitation, we have added the following statement regarding our observed underestimates extending to other regions (lines 256 – 260):
"Areas with similar characteristics to Hudson Bay may show similar biases in the estimated speed and direction of drift. This includes other seasonal systems (e.g., Baffin Bay), and those with slower drift (e.g., Kara and Laptev Seas) or without IABP buoys (see IABP, 2020 and Rampal et al., 2009 for coverage). Further, we observed the relative degree of bias increases with speed. If such scaling in bias exists in other areas, then the magnitude of underestimation may be greater in areas with faster speeds (e.g., Chukchi Sea)."

There are some minor comments that needs to be addressed before publication.
No response needed.

Row 44. What is meant with ecology? Ecological studies?
Yes - now reads "ecological research."

Row 87. 4h.
The Cryosphere formatting guidelines require spaces between number and unit.

Row 87. Are – > have a; Row 99. The – > the; Row 99. Replace – with for; Row 100. AMSRE – > AMSR-E
Changes made.

Row 156. Please explain what: intercept, CI, df, t and p stands for
Our methods explain that "we used an intercept-only [GLMM$_{PQL}$] . . . wherein a significant intercept represents a significant difference between the model and the collar speeds." We have defined 'CI' and 'df' and clarified 't' and 'p' throughout the manuscript. Parenthetical now reads (lines 159 - 160):

"(GLMM$_{PQL}$: intercept ± 95 % confidence interval (CI) = -3.0 ± 1.2 km d$_{-1}$, degrees of freedom (df) = 1657, t-value = -4.8, p-value > 0.0001; Figure 5)"

Row 181. What kind of biases were identified?
We have clarified the biases identified in the first discussion sentence, which now reads (lines 185 - 186): "Using drifting collars as reference data for validation, we identified biases in the estimated speed and direction of NSIDC modelled sea ice drift model."

Row 199. Remove "but see"
Change made.

Does TC allow for paper in review to be included in the reference list? If not please remove Klappstein et al. in review from the reference list.
This paper is now published, and its reference updated.

Row 257-258. Please provide the link to the data here.
The data link has been added. It is available at https://doi.org/10.7939/DVN/KUIZ7G

Row 266. Please provide the link to the data here.
Change made.

Row 326. What type of reference is this? PhD thesis? Report?
This is a BSc thesis. The citation now includes "BSc Thesis, Department of Physics Lund University"

[revised manuscript text omitted]